# Technologies and Equipment of Mechanized Blossom Thinning in Orchards: A Review

Xiaohui Lei [1], Quanchun Yuan [1], Tao Xyu [1], Yannan Qi [1], Jin Zeng [1], Kai Huang [1], Yuanhao Sun [1], Andreas Herbst [2] and Xiaolan Lyu [1,*]

1   Institute of Agricultural Facilities and Equipment, Jiangsu Academy of Agricultural Sciences/Key Laboratory of Modern Horticultural Equipment, Ministry of Agriculture and Rural Affairs, Nanjing 210014, China; leixiaohui@jaas.ac.cn (X.L.)
2   Institute for Chemical Application Technology of JKI, 38104 Braunschweig, Germany
*   Correspondence: lxlanny@126.com

**Abstract:** Orchard thinning can avoid biennial bearing and improve fruit quality, which is a necessary agronomic section in orchard management. The existing methods of artificial fruit thinning and chemical spraying are no longer suitable for the development of modern agriculture. With the continuous acceleration of the construction process of modern orchards, blossom thinning mechanization has become an inevitable trend in the development of the orchard flower and fruit management. Based on relevant reports in the past 20 years, the paper discusses the current level of development of mechanized blossom thinning technologies and equipment in orchards from three aspects: mechanism research, machine development, and intelligent upgrading. Firstly, for thinning mechanism research, three directions were investigated: the rope flexible hitting force, thinning agronomic requirements, and the fruit tree growth model between thinning and fruit yields. Secondly, for marketable machine developments, two types of machines were investigated: the hand-held thinner and tractor-mounted thinner. The hand-held thinner is mainly suitable for traditional old orchards with a messy canopy structure, especially in the interior and top of the canopy. The tractor-mounted thinner is mainly suitable for orchards with the same crown structure, such as the hedge type, trunk type, and V-type. Thirdly, for equipment intelligent upgrading, the research of the intelligent detection algorithm for inflorescence on the fruit tree was investigated, for species including the apple, pear, citrus, grape, litchi, mango, and apricot. Finally, combining the advantages and disadvantages of the research, the authors propose thoughts and prospects, which can provide a reference for the design and applications of orchard mechanized blossom thinning.

**Keywords:** agricultural machinery; orchard; thinning; mechanism; machine; intelligent



## 1. Introduction

China has an orchard area of over 12 million hectares and a total output of nearly 300 million tons, making it the largest fruit producer and consumer in the world [1–3]. With the continuous expansion of the orchard planting area and the increasing requirements for large-scale management, the importance of mechanized orchards is becoming increasingly prominent. However, the research on related technologies started late and had a poor foundation, resulting in an orchard comprehensive mechanization rate of less than 30% [4,5]. Especially in some agronomic sections with a high labor demand, high labor intensity, and tight agricultural seasons, mechanization has not been achieved. Moreover, China is facing an aging population of orchard farmers [6]. The lack of efficient and labor-saving agricultural equipment has become a bottleneck that restricts the development of the fruit industry.

Thinning can avoid biennial bearing and improve fruit quality, which is a necessary agronomic section in pear orchard management [7,8]. The existing methods of artificial fruit thinning and chemical spraying are no longer suitable for the development of modern

agriculture. Artificial fruit thinning consumes labor and time, wastes tree nutrients, and has a high-risk factor for tree-top operations [9]. The application of chemical thinning pollutes the environment, and the agents are easily affected by factors such as the spraying time, preparation concentration, and working site environment [10–12]. As a result, they are rarely used in actual field operations. Orchard thinning, as a key step in increasing orchard yields, still relies on manual operations. Manual operation is time-consuming and inefficient, and the experience requirements for agricultural operators in blossom thinning work are still high. With the continuous acceleration of the construction process of modern orchards, blossom thinning mechanization has become an inevitable trend in the development of orchard flower and fruit management.

Based on relevant reports in the past 20 years, the paper discusses the current level of development of mechanized blossom thinning technologies and equipment in orchards from three aspects: mechanism research, machine development, and intelligent upgrading. Combining the advantages and disadvantages of the research, the authors propose thoughts and prospects on orchard mechanized blossom thinning.

## 2. Mechanism Research

### 2.1. Thinning Force

The process of striking is complicated, which is the dynamic impact, friction, and vibration behavior of a flexible rope interacting with branches, inflorescences, and young fruits of fruit trees. It is difficult to estimate the impact force of the thinning rope on branches and inflorescence, and the damage degree of non-target parts such as branches and leaf buds is also difficult to estimate. Scholars have carried out relevant research on the flexible impact mechanism.

Hu et al. used a universal materials tester to measure the tensile and shear forces of apple branches' pedicel node and receptacles corolla node [13]. After repeated tests, the results showed that the average maximum tensile and shear force of the branches pedicel node in full bloom period were 3.5 N and 2.82 N, respectively. The average maximum tensile force that the receptacles corolla nodes can bear is 1.6 N. Combined with the ADAMS multi-body dynamics simulation, Hu's team also designed a performance test bench for the end actuator of the thinner. They tested the impact force of three kinds of thinning ropes made of a solid rubber strip, hollow vinyl strip, and nylon braid under different lengths, wire diameters, and speeds of the thinning shaft [14–18]. The optimal parameters of the actuator are as follows: the material of the thinning rope is solid rubber, the length is 20 cm, the linear diameter is 5 mm, the rotational speed of the spindle is 960 r/min, and the impact force of the thinning rope on the test branch is 5.39 N. Starting from the mechanical characteristics of peach branches, Yuan et al. obtained the stress curve of the branches and the shear strength of the branches is 6 MPa. Meanwhile, they established a rigid–flexible coupling multi-body dynamics model for the striking force of branches, analyzed the striking force of the thinning rope on branches at different rotational speeds, and concluded that the striking force range of the thinning rope is 3.3–7.6 N when the rotating speed of the spindle is 240–480 r/min [19–21]. Assirelli A. et al. studied the thinning power of young peach fruit, when the fruits were 20–40 mm in size and different angles were evaluated to simulate the various ways in which the thinner hits the fruits [22]. The analysis of the different angles showed that, on average, the fruits are detached more easily if the force is applied with a 90° angle, respective to the fruit position on the branch. On the other hand, if the force is applied with an angle of 0°, the average force required is three times higher.

### 2.2. Thinning Agronomy

The research of the thinning mechanism should not only solve the problem of flexible hitting, but should also solve the problem of thinning agronomy. Due to the different species of fruit trees, the number of flower buds contained in their inflorescence is different, and their agronomic requirements for thinning are also different. The apple, pear, and grape belong to a mixed inflorescence and there are multiple flower buds on one inflorescence,

while the peach and nectarine belong to a single inflorescence and there is only one flower bud on one inflorescence. Is the effect of thinning better in the bud stage or in the full bloom stage? Is removing the side flower better than the center flower when thinning the apple tree? What is the appropriate setting for the proportion of inflorescence thinning? Relevant scholars have conducted research on such thinning agronomic issues.

Sidhu et al. compared artificial bud extinction and blossom thinning in the 'Scilate' Apple; the results demonstrated that artificial bud extinction consistently outperformed blossom thinning in terms of an improved fruit set, return bloom, and fruit weight. The fruit quality parameters, such as flesh firmness, total soluble solids, dry matter content, malic acid content, and fruit shape, were also improved under the artificial bud extinction regime [23]. Oliveira et al. evaluate the effect of shoot heading and of thinning in different development stages of flowers and fruits on the fruit production and quality of the 'BRS Kampai' peach. The results showed that thinning during flowering and at the beginning of fruit growth increases the fruit size, and that shoot heading reduces plant production, but does not significantly increase the fruit size [24]. Szot et al. evaluate the effect of the crop load, thinning practices, and position in the tree crown on the quantity and quality of the 'Szampion' apple [25]. The thinning treatments were performed at the pink bud stage and full bloom stage, leaving only the king flower or lateral flower. The best results in terms of the regularity of yielding and high-quality fruits after thinning at the pink bud stage to the king flower were obtained. Han et al. studied the effects of berry thinning on bunch compactness, grape sugar accumulation, and subsequent wine quality in the Vitis vinifera L. Cabernet Sauvignon [26]. Based on the decreasing proportion of berries in one bunch, the treatment was designated as a 25% decrease and a 50% decrease. The results showed that the different berry thinning treatment lowered bunch compactness accordingly; the content of sugar, anthocyanins, total phenols, and the mass of mature berries were all significantly enhanced with a decreasing compactness at the same harvested time. Nie et al. studied the effects of blossom thinning on the fruit setting rate and fruit quality of apple inflorescences at different periods [27]. The test results showed that the thinning technique had a better fruit setting rate of apple inflorescences, fruit weight per fruit, color index, finish index, and soluble solid content than the traditional technique. Hua et al. took the Tainong No. 1 mango tree, aged 6 to 8 years, as the test material to study the optimal date of blossom thinning of the mango. The test results showed that the blossom thinning effect was better at the early bloom stage [28].

*2.3. Thinning Model*

In order to study the relationship between thinning and fruit yield, relevant scholars established the fruit tree growth model and theory. Iwanami et al. developed a theoretical model using the 'Fuji' apple to explain the relationships among the timing of thinning, crop load, fruit weight, and bloom return [29]. The rate of flower-bud formation in the current year could be explained by a regression model of the timing of the thinning, crop load, and rate of flower-bud formation in the previous year. The fruit weight in the current year could be explained by a regression model of the timing of the thinning, crop load, rate of flower-bud formation, and shoot length in the current year and the previous year. Pellerin et al. proposes that thinning is a partial transfer of potential flower buds from one year to the next year and estimates the maximum repeatable sequence of flower buds without biennial bearing [30]. Manfrini et al. investigate the feasibility of a spatial analysis in apple orchards to assist growers with decision making [31]. A variation in the spatial distribution of the fruit load prior to the thinning was observed, indicating a possibility to spatially and differentially manage the orchard. No spatial variation in the fruit number was observed prior to the harvest, indicating that thinning had removed the previously-observed spatial variation in the crop load. Reginato et al. assessed the relationship between crop load and fruit size or crop load and yield efficiency by a regression analysis for nectarines and cling peaches [32]. With this methodology, the predicted crop value can be established for different growing conditions leading to an improved crop load management. This

will permit growers to optimize the net return that can be obtained for a specific orchard. Understanding how crop value is affected by crop load for different cultivars can lead to better decisions in the design and establishment of new orchards.

## 3. Machine Development

### 3.1. Hand-Held Thinner

The hand-held thinner belongs to the semi-mechanized orchard management equipment, which is mainly composed of an energy supply device, operating lever, and thinning action actuator. According to the operating principle of the thinning action actuator, it can be divided into three kinds: an impact type, finger brush type, and vibration type. During the operation, the operator holds the operation lever and thins the target's inflorescences, or vibrates the canopy branches according to the agronomic requirements of the fruit tree to reduce the blossoms and fruits. The operator determines the percentage of inflorescences and young fruits that need to be removed by subjectively judging. Because of the small structure and portability, operators can directly carry the work in hand. The hand-held thinner is mainly suitable for traditional old orchards with a messy canopy structure, especially in the interior and top of the canopy.

The string type thinner is widely used, and there are many marketable products. It is controlled by a DC motor to control the speed of the spindle, so as to control the hitting force of the rope on the target inflorescence. The electric hand-held blossom thinner made by Infaco Co. Ltd. (Cahuzac sur Vère, France) has a rotary head with a five-finger comb and is powered by a 48 V electric motor; it was equipped with a portable battery bag which facilitated worker mobility in the field [33]. The orchard blossom thinner made by Cinch Co. Ltd. (Shelby Township, MI, USA) has two types of electric and manual for customers to choose; the installation hole on the spindle is reserved for the rope and is used to adjust the density of the rope according to the flower intensity. It is used on peach, cherry, apple, plum, apricot, and nectarine varietals [34]. The AF 100 electronic blossom/young fruit thinner made by Lakewoodproducts Co. Ltd. (Wellington, New Zealand) is particularly effective with stone fruit, such as nectarines, apricots, peaches, and plums. Its loop is made from high strength flexible rubber fitted on a shaft and the rubber loops will adapt to the location to remove the blossom without damaging the leaves or bark [35]. For the electric finger blossom/young fruit thinner, the rotation of the finger dial was controlled by the motor, and the target blossom/young fruit falls off in the friction and impact action with the finger brush. The Giulivo-plus electric finger thinner, made by Volpi Co. Ltd. (Casalromano, Italy), had a head with six rotating fingers and was powered by a 12 V electric motor; electricity was supplied by a 12 V, 75 Ah car battery, which remained on the ground, and a 15 m long electric extension cord [36]. For the shaking thinning device, the vibration was formed by the CAM mechanism action, acting on the branches of fruit trees, which is used for picking and thinning small fruit by acting on the branches of fruit trees. The hand-held shaker made by Campagnola Co. Ltd. in Italy is an air compressor which provided pressure between 1.0 and 1.2 MPa. The mobility of this device was limited due, in part, to the presence of the flexible hose that fed the compressed air to the shaker [37]. The marketable production of the hand-held thinner is shown in Table 1.

Relevant scientific research institutions have also designed their own hand-held thinners. Most of them are improved designs of market products and exist in the form of patents. Fruit varieties include citrus, apple, pear, peach, grape, kiwi, grape, etc. The example patent of the hand-held thinner is shown in Table 2 [38–47]. Researchers also carried out field tests on the performance of the hand-held thinners. Lei et al. developed a hand-held electric rotating rope blossom thinner for the 'Cuiguan' pear orchard: its spindle speed is from 0–900 r/min and extension rod length range is from 0.95 to 1.6 m. Compared with the hand thinning, the thinner can shorten the thinning time of the small canopy layer orchard by 30.71% and the Y-trellis orchard by 48.68%, respectively [48,49]. Wang et al. developed a hand-held mechanical thinning device suitable for the apple, stone fruits, and sweet cherry, with its spindle speed from 500 to 3000 r/min. Field test results indicated that

the test device could remove 61.1%, 30.8%, and 18.0% of flowers on a single branch with a swipe of around 0.5 m/s under high, medium, and low speed settings, respectively [50,51]. Martin et al. tested the Giulivo-plus electric finger thinner and the Campagnola hand-held shaker on a peach tree. The results indicated that the finger thinner reduced the time by 46% and the shaker reduced the time by 28%; two thinners reduced the crop load by 38% and increased the mass of the fruit by 47% at harvest compared to non-thinned trees [52]. After that, Martin et al. developed a hand-held fruit thinner prototype: it had a rotating cylinder with 10 flexible cords, placed at the top of a pole 2 m in length. They carried out a field test with the Infaco and Giulivo-plus thinner in peach orchards; there is no differences among them in terms of thinning time and the number of fruits per cm$^2$ of the trunk cross-sectional area at harvesting. Hand thinning took 385 h/ha, and mechanical thinning reduced this time by 89%. The cost of hand thinning was 4.8 €/tree, whereas the cost of mechanical thinning ranged from 0.4 to 1.1 €/tree [53,54]. Spornberger et al. tested the portable thinner and manual thinning in organically managed cherry orchards at the stage of a pea size; the portable thinner showed a high thinning effect, and because of lower costs, it is more suitable for farmers than the hand thinning of flowers or fruits [55].

**Table 1.** Marketable productions of the hand-held thinner.

| Brand | External Structure | Working Site | Main Performance Parameter |
|---|---|---|---|
| Infaco of Infaco Co. Ltd. in France | | | Weight: 2.9 kg<br><br>Range of extension rod: 1.5 to 1.9 m<br><br>Range of spindle rotation speed: 320 to 1900 r/min |
| Cinch of Cinch Co. Ltd. in the USA | | | Weight: 3 kg<br><br>Length: 2 m<br><br>Range of spindle rotation speed: 0 to 2000 r/min |
| Lakewoodproducts of Lakewoodproducts Co. Ltd. in New Zealand | | | Weight: 0.5 kg<br><br>Length: 0.3 m<br><br>Range of spindle rotation speed: 0 to 1500 r/min |
| Volpi of Volpi Co. Ltd. in Italy | | | Weight: 2 kg<br><br>Range of extension rod: 2.1 to 3.6 m<br><br>Range of spindle rotation speed: 714 to 833 r/min |
| Campagnolaof Campagnola Co. Ltd. in Italy | | | Weight: 1.9 kg<br><br>Range of vibratory frequency: 10 to 14 Hz |

**Table 2.** Patent of the hand-held thinner.

| Name | External Structure Drawing | Main Structure and Performance |
| --- | --- | --- |
| Hand-held kiwi fruit thinning device designed by Chinese Academy of Sciences | | It includes a telescopic rod, upper/lower blade clip, and three-tooth silicone sleeve. On the basis of improving the efficiency of thinning, more central flowers can be retained to prevent scratching young fruit [38]. |
| Electric knapsack blossom thinning machine of China Agricultural University | | It includes a back frame, telescopic rod mechanism, articulated mechanism, and adjustable tool head. The vertical height and spindle angle can be adjust by the operator and it has a good profiling effect [39]. |
| Hand-held pomelo blossom thinning device of Duwei Xianxi Fruit Industry Professional Cooperative in China | | It includes a hand-held rod, fixed rod, locking block, adjusting rod, and thinning ring. During operation, stick the thinning ring into the branch, and move it up and down with a hand pole to peel the blossom off the branch [40]. |
| Hand-held bag-shaped flower and fruit thinning device of China Agricultural University | | It includes a power battery, handle, bag thinning device, motor, U-type frame, fixed tool holder, and cutting blade. During operation, the U-type frame encloses the inflorescence and the motor drives the cutting blade rotation for thinning [41]. |
| Hand-held citrus flower thinning machine of Country Garden Co. Ltd. in China | | It includes a handle, power cord, battery, backpack, and movable shaft connecting rod. The rotating shaft of the device can be adjusted arbitrarily to the required operating angle [42]. |
| Electric flower and fruit thinning device of Mengxian Automation Equipment Co. Ltd. in China | | It includes a shell, thinning sleeve, driving motor, driving wheel, and cutting knife. During operation, the reserved blossoms are placed in the thinning sleeve, and the motor is driven to rotate and drive the cutting knife to remove the excess blossom [43]. |
| Device for thinning flower, fruit and small grains fruit | | It includes a fixed blade, moving blade, scissor amplitude regulator and eccentric wheel. During operation, press the switch with your fingers, and align the scissor with the target inflorescence [44]. |

**Table 2.** *Cont.*

| Name | External Structure Drawing | Main Structure and Performance |
| --- | --- | --- |
| Hand-held citrus flower thinning machine of Yang Fengsheng citrus Professional Cooperative in China | | It includes a fixed cylinder, telescopic rod, fixed plate, rack, limit plate, lifting block, moving block, rotating shaft, and a rotating gear. The mechanism can rotate while moving up and down to meet the needs of the working site [45]. |
| Ultrasonic target detection orchard thinning machine of South China Agricultural University | | It includes a work frame, spindle, thinning strip group, ultrasonic sensor, motor, angle adjusting mechanism, telescopic rod, and control mechanism. During operation, the ultrasonic detector performs a start–stop action of spindle according to the presence or absence of the inflorescence target [46,47]. |

*3.2. Tractor-Mounted Thinner*

The tractor-mounted thinner belongs to the mechanized blossom and fruit management equipment. The whole machine is operated by the tractor and the rotating power of the spindle is provided by the tractor hydraulic mechanism. The marketable productions are divided into three kinds according to the structure of the thinning arm: the single spindle, multiple spindles, and horizontal rotary spindle. The working mechanism between the thinning rope and the inflorescence is that of the hitting, rubbing, and vibrating. According to the fruit species, crown type, and other agronomic parameters, operators adjust the tractor advancing speed, the spindle operation angle, the spindle rotational speed, and the density of the thinning rope, so as to reasonably select the thinning intensity. The tractor-mounted thinner is mainly suitable for orchards with the same crown structure, such as the hedge type, trunk type, and V-type.

At present, the Darwin series orchard single spindle string blossom thinner made by Fruit Tec Co., Ltd. is the mainstream one in the market. The production is shown in Table 3 [56]. The Darwin S is mounted on the front of the tractor, so a better view of the spindle is ensured and thus, the operator can guide the spindle more easily on the tree canopy. The spindle rotational speed can be comfortably and continuously adjusted with buttons on the control unit in the driver's cabin to be optimally adapted to the driving speed. The Darwin PT is attached to a front loader directly and is designed to work mainly in the V-type canopy tree. The spindle has a tilt angle of 180°; this allows it to be lifted to the height of the tree-tops and work horizontally over the trees. If the vase trees are very wide, there is also the possibility of tilting the spindle into the tree's interior. The Darwin SmaArt replaces the subjective estimation of the blooming strength with the eye with objective detection by camera. To do this, a camera in front of the thinning spindle detects the blossom density of each individual tree and passes on the data to the on-board computer in real time. Using a thinning algorithm, the computer then calculates the optimum spindle speed and controls the thinning unit. As an option, the system can be combined with a GPS receiver. Using the GPS system, it is possible to detect each individual tree and to assign the data, such as the number of blossoms and the spindle speed, to the tree and to compare it later with the yield data.

**Table 3.** Darwin series single spindle string blossom thinner.

| Name | Working Site | Main Performance Parameter |
|---|---|---|
| Darwin S |  | Type: 150, 200, 230, 250, 300<br>Weight: 87 to 101 kg<br>Working height: 1.475 to 2.85 m<br>Driving speed: 6 to 18 km/h<br>Range of spindle rotation speed: 150 to 450 r/min<br>Working efficiency: 1.5 to 2.5 ha/h<br>Orchard species: apple, peach<br>Suitable canopy type: hedge wall |
| Darwin PT |  | Type: 250<br>Weight: 246 kg<br>Working height: 2.395 m<br>Driving speed: 6 to 18 km/h<br>Range of spindle rotation speed: 150 to 450 r/min<br>Range of rotation angle: 0 to 180°<br>Working efficiency: 1.5 to 2.5 ha/h<br>Orchard species: apple, peach<br>Suitable canopy type: hedge wall, V-type, open center |
| Darwin SmaArt |  | Type: 150, 200, 230, 250, 300<br>Weight: 130 to 152 kg<br>Working height: 1.475 to 2.85 m<br>Driving speed: 6 to 18 km/h<br>Range of spindle rotation speed: 150 to 450 r/min<br>Working efficiency: 1.5 to 2.5 ha/h<br>Orchard species: apple, peach<br>Suitable canopy type: hedge wall |

In recent years, scholars have carried out a large number of tests in apple, peach, and plum orchards with the Darwin series thinner. The range of the optimal working speed is 6 to 18 km/h, and the range of the spindle rotation speed is 150 to 450 r/min. Accurate working parameters need to be obtained in the field according to the fruit tree species, the shape of the canopy, and the period of blossom thinning. Wallis et al. evaluated the risk of fire blight development and spread after Darwin 300 blossom thinning in apple orchards. The results demonstrate that the use of the thinner should, therefore, be limited to orchards with no history of disease in the last 3 years and on days when predicted weather is not suitable for tree infection; there is a low risk for fire blight development and spread by mechanical thinning under an early blossom stage, especially when paired with a subsequent bactericide application [57,58]. Penzel et al. translated the Darwin 250 spindle rotational frequency to average kinetic energy. At a high flower set, thinning treatments of 0.23 J and 0.33 J were adequate settings to reduce crop load in 'Elstar' and 0.33 J in 'Gala' without yield loss [59]. Lordan et al. evaluated the working performance of Darwin 250 in 'Gala', 'Golden Delicious', and 'Fuji' apple orchards; two prediction models were developed to adjust the right tractor and spindle rotational speeds depending on the initial number of flower clusters [60]. Theron et al. evaluated chemical thinning with the Darwin thinner during the plum blossom stage; the method increased the fruit drop, fruit size, and fruit weight [61]. Cline et al. tested the Darwin 300 and hand thinning in apple and peach orchards; the result showed that mechanical thinning represents a viable method for initial crop load reduction, coupled with hand thinning after fruit, set to reach a final optimal production [62,63]. Baugher et al. tested the Darwin 300 in V-shaped and open-center trained peach orchards; they demonstrated that the mechanical thinner reduces labor requirements and improves fruit size [64–68].

Other companies also make related productions, such as the Eclairvale series orchard thinner (as shown in Table 4) made by La Canne Vale Co., Ltd. (Narbonne, France) [69]. The machine can be adopted to be both semi-mounted and mounted, to perform the thinning operation adequately: the total mass of the tractor with or without front ballast must guarantee the stability of a 3 m rear overhang. It has a freely rotating rotor onto which semi-rigid rods are attached. Its rotation occurs through the penetration of the rods into the canopy and the advancement of the tractor, which induces a slow rotation and rubbing of the rods against the branches, which causes some of the blossom or green fruit to fall. Rods are made of flexible glass fibers with a soft plastic end cap; it is the only wearing part and each rod can be easily replaced in less than one minute. Assirelli et al. tested the Eclairvale thinner in apricot and peach orchards [70–72]. In the apricot, the machine removed 20.8% of flowers and 43.6% of fruit, allowing 48% time-saving in the follow-up fruit manual thinning as compared with the hand thinning. In the peach, mechanical thinning at blooming time removed 63% of flowers, allowing 42.4% time-saving in the follow-up fruit manual thinning as compared with the hand thinning. Fruit damages always remained below the economic thresholds to marketable production or to the plant.

**Table 4.** Eclairvale thinner.

| Name | Working Site | Main Performance Parameter |
|---|---|---|
| Eclairvale EH | 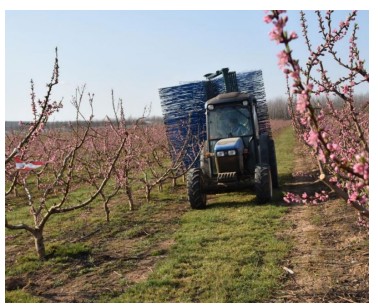 | Type: 1600, 2100, 2500<br>Weight: 580 to 1095 kg<br>Working height: 1.6 to 2.5 m<br>Driving speed: 1.5 to 15 km/h<br>Working efficiency: 0.25 to 3 ha/h<br>Orchard species: peach, nectarine, apricot, plum, organic apple, almond, pistachio<br>Suitable canopy type: cone, goblet, hedge wall |
| Eclairvale FR | | Type: 1600, 2100, 2500, 3000, 3500<br>Weight: 540 to 1325 kg<br>Working height: 1.6 to 3.5 m<br>Driving speed: 1.5 to 15 km/h<br>Working efficiency: 0.25 to 3 ha/h<br>Orchard species: peach, nectarine, apricot, plum, organic apple, almond, pistachio<br>Suitable canopy type: cone, goblet, hedge wall |

The FLEXITREE three-arm blossom thinner made by Clemens Co. Ltd. (Wittlich, Germany), is shown in Table 5, which is linked in front of a tractor [73]. Three arms are installed flexibly in different positions of the vertical rod and its structure can be changed by adjusting, which has the advantage of more penetration into the canopy. The device is flexibly configurable and can be perfectly adapted to the crown structure and different tree heights by means of various adjustment options. Lei et al. designed a three-arm tractor-mounted flower thinner named TTBT-300 for 'Y' trellis and trunk-type pear orchards; the field test showed that the thinner can save at least 60% of artificial fruit thinning time and the profitable area was 0.58 hm$^2$ [74,75]. Blanke et al. designed a three-arm tractor-mounted flower thinner named Bonner for plums and apples. In the plum orchard, the yield of Class one fruits increased per tree from 47% in the un-thinned controls up to 69%; the fruit mass was enlarged from 28 g in the un-thinned control to 30–32 g [76,77]. In the apple orchard, the portion of Class one fruits bigger than 70 mm was increased by 10% without yield loss and it reduced the subsequent hand thinning by, respectively, 32.5% [78,79]. They also

proposed an integrated coefficient of thinning (ICT) to develop the critical parameters of the machine. The optimum values are 10–40, where an ICT > 50 led to tree damage and ICT < 8 led to sub-optimum thinning efficacy [80,81].

**Table 5.** Three-arm blossom thinner.

| Brand | Working Site | Main Performance Parameter |
|---|---|---|
| Flexitree of Clemens Co. Ltd. | 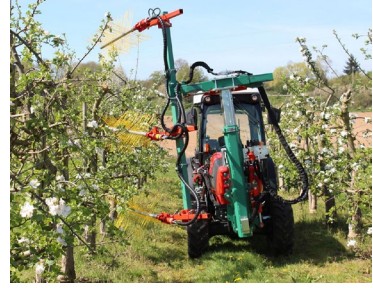 | Weight: 155 kg<br><br>Dimensions: 0.46 × 2.2 × 2.45 m<br><br>Lifting height: 0.95 m<br><br>Lateral inclination: inwards 18°, outwards 26°<br><br>Orchard species: apple, peach<br><br>Suitable canopy type: trunk type, Y-trellis, hedge wall |
| TTBT-300 designed by Jiangsu Academy of Agricultural Sciences | 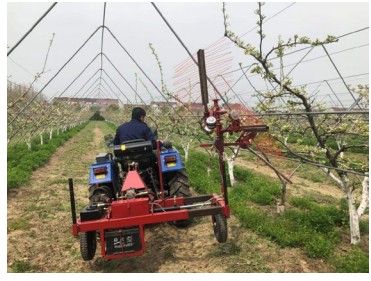 | Weight: 200 kg<br>Dimensions: 0.9 × 1.2 × 2.7 m<br>Arm extension range: 0–0.5 m<br>Spindle length: 1.1 m<br>Spindle rotation speed range: 0–300 r/min<br>Orchard species: apple, peach, pear<br>Suitable canopy type: trunk type, Y-trellis |
| Bonner designed by University of Bonn | 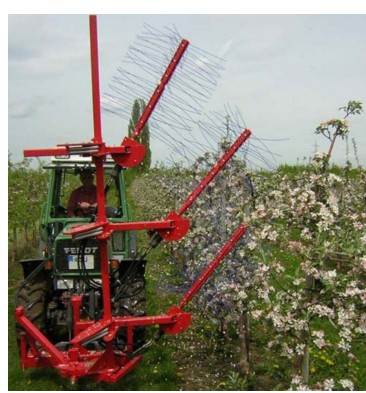 | Height: 3 m<br><br>Arm extension range: 0–0.5 m<br><br>Spindle rotation speed range: 0–500 r/min<br><br>Orchard species: apple, peach<br><br>Suitable canopy type: trunk type, Y-trellis, hedge wall |

The spiked-drum shaker made by the United States Department of Agriculture consisted of several panels of radially spaced nylon rods bolted to plates on a central spindle, as shown in Table 6. Schupp et al. evaluated the double spiked-drum shaker on the peach; it reduced peach crop load by an average of 36%, decreased follow-up hand thinning time by 20% to 42%, and increased fruit in higher market value size categories by 35% [82]. Miller et al. evaluated the single and double spiked-drum shaker on the peach to reduce the cost and time required for hand thinning peaches. At the 60% full blossom stage, the double-spiked drum shaker reduced crop load by 27% and the single spiked-drum shaker reduced crop load by 9%; they removed an average of 37% of the green fruit [83].

**Table 6.** Spiked-drum shaker.

| Name | Working Site | Main Performance Parameter |
| --- | --- | --- |
| Double spiked-drum shaker | 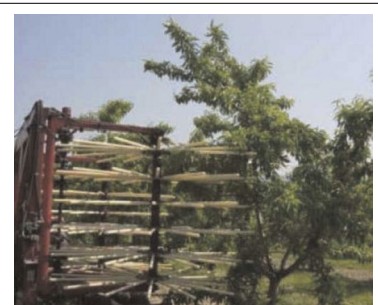 | Rotating drum diameter: 2.4 m<br>Rotating drum height: 1.5 m<br>Layer number of nylon rods: 6<br>Nylon rods number of one layer: 12<br>Nylon rods line diameter: 3.2 cm<br>Nylon rods length: 1.1 m<br>Orchard species: peach<br>Suitable canopy type: V-type |
| Single spiked-drum shaker | 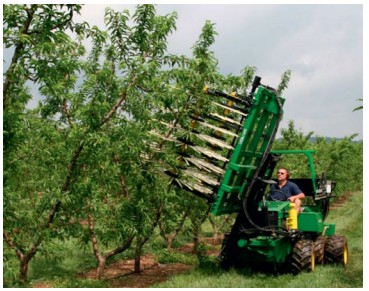 | Rotating drum diameter: 1.35 m<br>Rotating drum height: 1.2 m<br>Layer number of nylon rods: 8<br>Nylon rods number of one layer: 24<br>Nylon rods line diameter: 1.6 cm<br>Nylon rods length: 0.61 m<br>Adjustable tilt angle: 45°<br>Orchard species: peach<br>Suitable canopy type: V-type |

Other innovative thinners are shown in Table 7, and they belong to physical impact thinning. The one-rotor orchard horizontal rotary thinner made by Phil Brown Welding Co. Ltd. can be carried on a standard forklift tractor [84]. The rotor was driven by a hydraulic with a variable speed, and the ropes can be easily replaced when worn. Aasted et al. developed a system using an LIDAR to sense the canopy and automatically control the position of a modified Darwin string thinner position to maintain engagement [85]. They found that the laser control system performed similarly to the joystick control for the blossom counts. Though the laser system was slightly less engaged on the lower canopy, the overall scaffold performance was much closer. Lyons et al. developed a visual blossom thinning system [86]. It consisted of kinematic targeting and heuristic programming, a robotic arm, and a pomologically designed end-effector. The robotic arm had a consistent range of −1.26 cm to +1.57 cm vector magnitude per target location, and the end-effector brushes had a consistent range of −2.97 cm to +3.04 cm per target location. Li et al. developed a profiling control system for the litchi blossom thinner; the spatial position of the spindle could be adjusted by the translational screw and profile adjusting screw. The test results show that the dynamic mean errors of the translational screw and profile adjusting screw were 0.17 and 0.07 cm, respectively; the actuators have good position-adjusting accuracy and the proposed profiling control system can meet the requirements of real-time control [87,88]. Wouters et al. established a set of multi-spectral camera systems for the detection of pear inflorescence, which can identify pear inflorescence in six bands of visible and near infrared spectra [89]. The test results show that approximately 87% of the visible floral buds were detected correctly with a low false discovery rate (<16%).

**Table 7.** Other innovative thinners.

| Name | Working Site | Main Performance Parameter |
|---|---|---|
| One-rotor orchard horizontal rotary thinner of Phil Brown Welding Co. Ltd. | 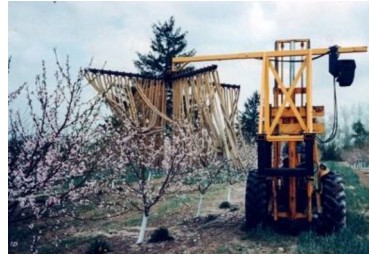 | Hydraulic driven rotor with variable speed, adjustable for different tree row widths, and rubber ropes can be easily replaced when worn.<br>Orchard species: peach<br>Suitable canopy type: open center |
| Darwin thinner with LIDAR scanning system of Carnegie Mellon University | 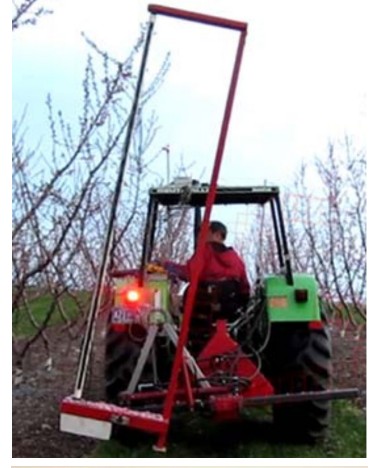 | Tractor speed: 1.6 km/h<br>Spindle rotation speed: 240 r/min<br>Spindle lateral control time: 5 s<br>Spindle angle control time: 4 s<br>Orchard species: peach, apple<br>Suitable canopy type: V-type, hedge wall |
| Visual blossom thinning system of Pennsylvania State University | 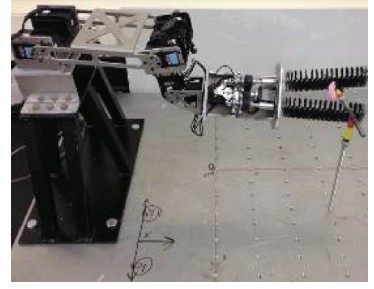 | Robotic arm precision: −1.26 cm to +1.57 cm<br>End-effector brushes precision: −2.97 cm to +3.04 cm<br>Orchard species: peach |
| Litchi blossom thinner profiling control system of South China Agricultural University | 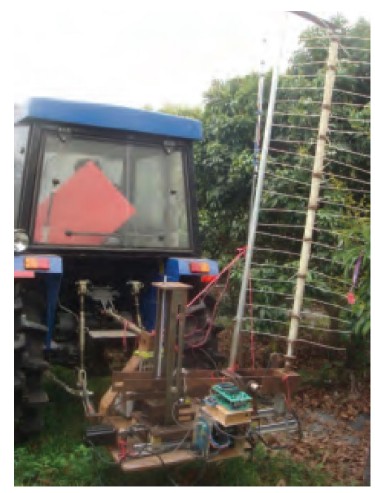 | Dynamic mean errors of translational screw: 0.17 cm<br>Dynamic mean errors of profile adjusting screw: 0.07 cm<br>Orchard species: litchi |
| Multispectral camera system for pear inflorescence designed by Catholic University of Leuven | 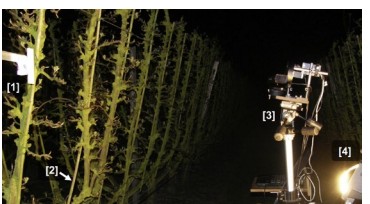 | Detection precision: 87%<br>False detection rate: <16%<br>Orchard species: pear |

In the patent, orchard blossom thinners are shown in Table 8. There are mainly four kinds: the single spindle string thinner has an angle adjusting device, multi-arm profiling thinner, multi-sensors automatic thinner, and machine vision intelligent thinner. The blossom thinner angle adjusting device is designed for the single spindle string thinner, aiming at adjusting the spindle angle to adapt to different tree canopies. The multi-arm profiling thinner is a pure mechanism innovation model, which realizes fruit tree canopy profiling through mechanism deformation. The multi-sensors automatic thinner adds sensors such as the radar, infrared probe, and ultrasonic probe to the existing blossom thinning equipment to realize the operation of the spindle angle and rotation speed adjusting. The machine vision intelligent thinner is a kind of intelligent blossom thinning equipment, which replaces human eyes with a high-definition camera, replaces the human brain with a deep learning convolutional neural network, and replaces the human hand with a mechanical flower thinning arm. At present, most of the patented products are a conceptual design, which has a certain theoretical basis. Some prototypes have been successfully developed, but there is still a certain distance to commercialization.

**Table 8.** Example patent of the tractor-mounted thinner.

| Name | External Structure Drawing | Main Structure and Performance |
|---|---|---|
| Tractive blossom and fruit thinning device designed by Xinjiang Academy of Agricultural and Reclamation Sciences | 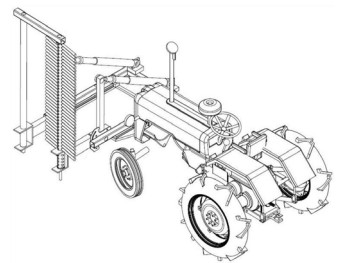 | It includes a U-type bracket, bottom mounting frame, L type connecting frame, and flower assembly. During operation, the spindle angle can be adjusted, which can adapt to the working environment of different terrains and canopies [90]. |
| Orchard cruise blossom thinning device of South China Agricultural University | 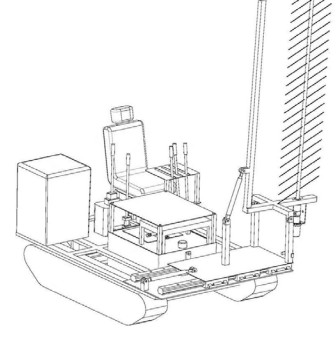 | It includes a blossom thinning mechanism, cruise mechanism, control system, and hydraulic system. When working, the distance sensor transmits the signal to the control system, and the control system controls the hydraulic system according to the target distance information, so that the hydraulic system drives the blossom thinning mechanism and the cruise mechanism [91]. |
| Blossom thinner of Qingdao Agricultural University | 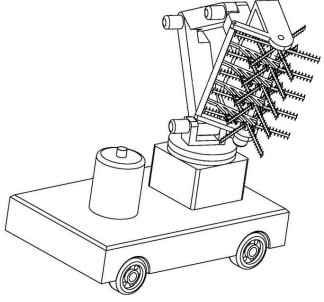 | It includes a trolley body with wheels, a supporting arm, and a flower thinning mechanism. It can ensure that the flower thinning agent can be completely sprayed on every part of the tree canopy [92]. |

**Table 8.** *Cont.*

| Name | External Structure Drawing | Main Structure and Performance |
| --- | --- | --- |
| Litchi profiling blossom mechanism of South China Agricultural University | 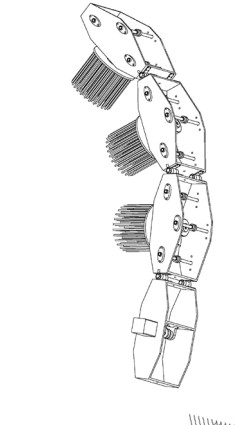 | It includes a base bracket, support bracket, spring articulated mechanism, and blossom thinning mechanism. During operation, the profiling function is realized through the coordination of a hinged structure with a torsion spring, support bracket, motor, and wire rope [93]. |
| Multi-segment arm profiling blossom and fruit thinning device designed by Jiangsu Academy of Agricultural Sciences | 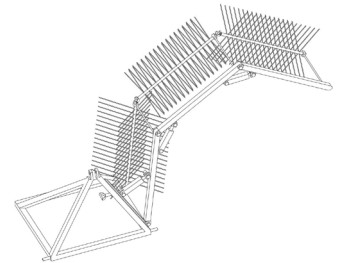 | It includes a suspension frame, transmission shaft, support bracket, hydraulic cylinder, thinning arm, and brush. During operation, the thinner is hung behind the tractor, which is used in orchards of the different tree canopies [94]. |
| Electric orchard blossom thinner designed by Jiangsu Academy of Agricultural Sciences | 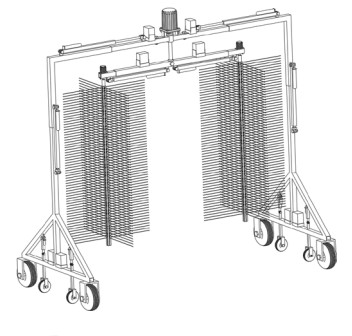 | It includes adjustable support, a thinning device, and a walking device. During operation, the height, width, and distance between the two spindles can be adjusted to avoid the damage of fruit trees in the process of flower-thinning, so that the thinner can adapt to fruit trees of different growth periods [95]. |
| Adaptive blossom thinner of South China Agricultural University | 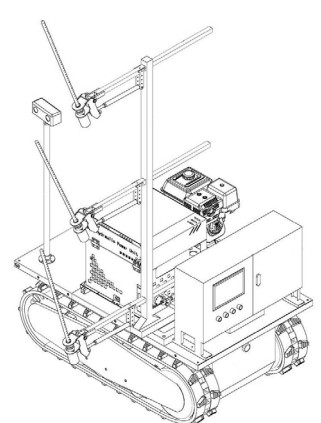 | It includes a carrier, multi-stage adaptive arm, intelligent inflorescence density identification system, adaptive control system, and hydraulic drive station. During operation, the image of fruit tree inflorescence is collected by the camera; the position and density information of blossoms are obtained by the intelligent inflorescence density recognition system, and the action of the thinning arm is controlled [96]. |

**Table 8.** *Cont.*

| Name | External Structure Drawing | Main Structure and Performance |
|---|---|---|
| Wide row dense planting orchard intelligent blossom thinner of Hebei Agricultural University | 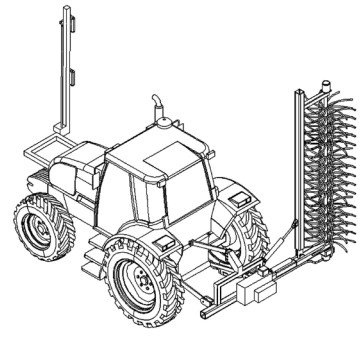 | It includes an unmanned tractor, camera, fixed bracket, and thinning device. During operation, the angle, distance, and speed of flexible brushing can be adjusted [21]. |
| Y trellis pear orchard profiling blossom thinner of Jiangsu Academy of Agricultural Sciences | 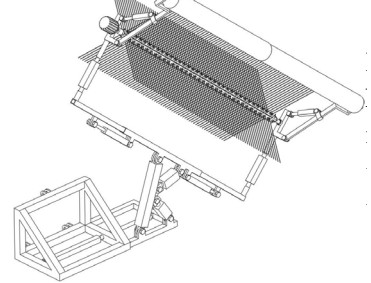 | It includes a suspension frame, holding bracket, hydraulic cylinder, thinning spindle, and distance roller. During operation, the angle and height of the telescopic thinning spindle can be adjusted to match the different sizes of the Y trellis [97]. |
| Blossom thinning arm based on image recognition of Shandong Agricultural University | 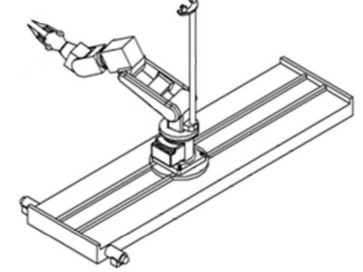 | It includes a controller, image acquisition equipment, mechanical arm, and actuator. During the operation, the image acquisition device collects the inflorescence image, the operator selects the inflorescence to be cut off, and the controller controls the mechanical arm and actuator to complete the thinning operation [98]. |
| Electromagnetic blossom thinning device for apple tree of HeXi University | 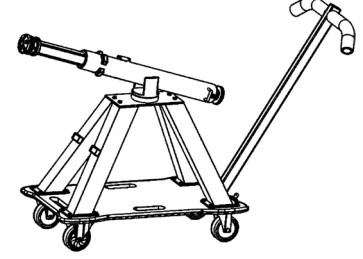 | It includes a support frame, turntable, connecting plate, tool head, telescopic rod, CCD camera, and control circuit. During the operation, the CCD camera collects the inflorescence image, the background computer obtains the three-dimensional coordinate information of the target object, and the embedded computer controls the motor to control the telescopic rod to move and locate the target object [99]. |
| Double roller intelligent blossom thinner based on machine vision of Shandong Agricultural University | 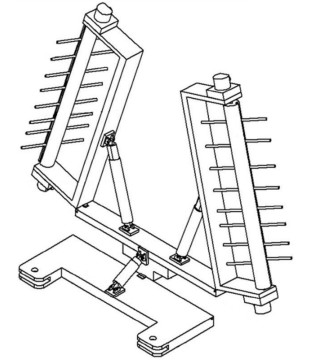 | It includes a fixed base, support platform, image acquisition device, and thinning device. During operation, the image acquisition equipment collects the height, canopy shape, position, and blossom density information of the fruit tree, and the controller determines the inclination angle and rotation speed of the roller [100]. |

## 4. Intelligent Upgrading

With the maturity and development of technologies such as artificial intelligence, 5G, the internet of things, and blockchain, agriculture will usher in a new period of development. The development of intelligent agricultural machinery and equipment has become a focus of the industry, especially the innovative application of machine vision intelligence technology represented by deep learning in the field of agricultural mechanization. The number and position of inflorescence among individual fruit trees vary, and fixed machine operation parameters cannot ensure the consistency of flower cluster density among different tree bodies. The excessive removal of inflorescence will reduce the yield of fruit trees, while too low of a removal will increase the subsequent workload of manual fruit selection. The machine operation method based on human eye evaluation cannot solve the above problems very well. In the research of intelligent detection algorithms for inflorescence on fruit trees, relevant researchers have made certain attempts.

In the field of deep learning inflorescence recognition, the apple has been the main research target. Researchers have compared the accuracy and detection time of various models for detecting the apple's flower intensity and growth status. Chen et al. proposed an apple inflorescence recognition algorithm based on Yolov5s, which incorporated a cooperative attention mechanism, bottleneck module, and CIOU loss function [101]. The model achieved an mAP value of 94.90%, which improved upon Faster R-CNN, Yolov3, Yolov4, and Yolov5 by 1.98%, 7.1%, 5.42%, and 2.53%, respectively. Shang et al. used machine vision and Yolov5s to detect apple flowers under different weather, colors, and light conditions [102]. The model achieved an mAP increase of 8.15%, 9.75%, and 9.68% compared to the Yolov4, SSD, and Faster-RCNN models, respectively. Additionally, the model size decreased by 94.23%, 84.54%, and 86.97%, respectively, and the detection speed increased by 126.71%, 32.30%, and 311.28%, respectively. Wang et al. proposed a phenology distribution estimation method named DeepPhenology for apple flowers based on CNNs using RGB images [103]. The model efficiently mapped the flower distribution on an image-level, row-level, and block-level, achieving an average Kullback–Leibler divergence value of 0.23 over all validation sets and an average KL value of 0.27 over all test sets. Tian et al. proposed an instance segmentation model for detecting and segmenting apple flowers with three different levels of growth status: bud, semi-open, and fully open [104]. The model improved upon Mask Scoring ReCNN with a U-Net backbone, achieving a precision of 96.43%, recall of 95.37%, F1 score of 95.90%, mean average precision of 0.594, and mean intersection over union of 91.55%. Farjon et al. proposed an estimator for apple blooming levels based on Faster R-CNN, achieving a high agreement level (0.78–0.93) between the algorithm blooming estimation and human judgments of several experts [105]. Wu et al. proposed a deep learning algorithm for the real-time and accurate detection of apple flowers based on the channel pruned Yolov4 [106]. The model achieved an mAP of 97.31%, higher than Faster R-CNN, Tiny-Yolov2, Yolov3, SSD 300, and EfficientDet-D0 algorithms by 12.21%, 15.56%, 14.19%, 5.67%, and 7.79%, respectively. Table 9 shows the performance of each algorithm. There have been several studies conducted by researchers to detect apple inflorescence using image processing methods. For instance, Zhang et al. proposed a novel strategy that combined UAV and ground-based RGB image data to detect flowering intensities in a Dutch Elstar apple orchard [107]. Their approach yielded a $R^2$ of 0.7 and a RMSE lower than 20, indicating a high correlation between the image-derived flower index, the white index, and the in-field counts of the cluster number. Liu et al. developed an accurate estimation model for apple fuzzy clustering with a complex background, which showed a high reliability and applicability [108]. The model estimates flower quantity quickly and accurately with an overall accuracy rate of 92%. Črtomir et al. developed an apple yield forecast hybrid model based on artificial neural networks [109]. The forecast of the hybrid method showed a higher accuracy than an image analysis. Aggelopoulou et al. developed an image processing-based algorithm that predicts tree yield by analyzing the picture of the tree at full blossom [110]. Their test results indicated that the potential yield could be predicted early in the season from flowering distribution maps, and could be used

for orchard management during the growing season. Overall, these studies demonstrate the potential of image processing methods for detecting and predicting apple inflorescence, which could be beneficial for orchard management and improving yields.

**Table 9.** Apple inflorescence recognition and detection.

| Training Model/Method | Platform | Image | Performance Index | Reference |
|---|---|---|---|---|
| Yolov5 | Software platform: Windows10, PyCharm2022, Pytorch1.7, Python 3.8. Hardware platform: Intel(R)Core(TM)i7-10700F, CPU2.9GHz, NVIDIA GeForce RTX3060. |  | Precision: 98.07% <br><br> Recall: 97.56% <br><br> mAP: 94.9% <br><br> Detection speed: 77 FPS | [101] |
| Yolov5s | Software platform: Windows10, PyCharm2022, Pytorch1.6, Python 3.8. Hardware platform: Intel(R)Core(TM) i7-10700K, CPU3.80 GHz, NVIDIA GeForce RTX2080Ti. |  | Precision: 87.7% <br> Recall: 94% <br> mAP: 97.2% <br> Trained model size: 14.09 MB <br> Detection speed: 60 FPS | [102] |
| Vgg16 | Software platform: Ubuntu 18.04 LTS, Pytorch1.1. Hardware platform: Intel(R)Core(TM) i9-9900KF, CPU3.60 GHz, NVIDIA GeForce RTX2080Ti. |  | Kullback-Leibler divergence value: 0.23 | [103] |
| Mask R-CNN | Hardware platform:NVIDIA Tesla V100 server. |  | Precision: 96.43% <br> Recall: 95.37% <br> F1 score: 95.90% <br> mAP: 59.4% <br> mIoU: 91.55% | [104] |
| Vgg16 | Software platform: Matlab 2014a. |  | Precision: 68.3% <br><br> Recall: 70% | [105] |
| Yolov4 | Software platform: Python 3.8, Microsoft Visual Studio 2015. Hardware platform: Intel(R)Core(TM) E5-1620, CPU3.50 GHz, NVIDIA GeForce RTX2080Ti. |  | mAP: 97.31% <br><br> Trained model size: 12.46 MB <br><br> Detection speed: 72 FPS | [106] |

**Table 9.** *Cont.*

| Training Model/Method | Platform | Image | Performance Index | Reference |
|---|---|---|---|---|
| Point cloud with image colour | Unmanned aerial vehicle, RGB camera, real time kinematic positioning system |  | $R^2$: >0.65 RRMSE: <20% P-stat: <0.005 | [107] |
| Fuzzy cluster | RGB camera |  | Precision: 92% | [108] |
| Picture texture | Digital camera |  | Precision: 82% | [110] |

In addition to the apple, researchers have also conducted studies on inflorescence detection and classification in other fruit trees such as pear, citrus, grape, litchi, mango, and apricot. For example, Xia et al. proposed a pear inflorescence recognition algorithm Ghost-Yolov5s-BiFPN based on Yolov5s to address the problem of inflorescence detection and classification in intelligent pear orchard production [111]. Their field test results showed that the mAP and recall rate were improved by 4.2% and 2.7%, respectively, compared with the original Yolov5s network, and the detection time and model parameters were reduced by 9 ms and 46.63%, respectively. Zhou et al. proposed a density classification of the pear flower images method based on the improved density peak clustering algorithm; it adopted the soft statistics method for density calculation, which had continuity and was more accurate for density classification [112]. Lyu et al. proposed a lightweight citrus recognition model using cascade fusion Yolov4-CF, which achieved a frame rate of 30 FPS on the FPGA side and could meet the demands of real-time monitoring for florescence information [113]. Deng et al. proposed an instance segmentation algorithm, in which the mask-RCNN can simplify the relatively complex object segmentation by simple detection [114]. Du et al. proposed a new method to locate the clamping points of fruit stems for table grape thinning based on an improved Mask-CNN [115]. Their test results showed that the location accuracy was 90% and the total time was 0.3 s, and the maximum location error in x, y, and the total location error were 10, 12, and 16 pixels, respectively. Lin et al. proposed a method using multicolumn-CNN to calculate the number of litchi blossoms by generating a density map [116]. Their approach achieved a mean absolute error of 16.29 and a mean square error of 25.40, which outperformed traditional counting methods using target detection. Wang et al. presented two machine vision methods for assessing the flowering intensity of mango orchards [117]. The correlation coefficient between their methods and human visual counts ranged from 0.78, indicating a strong

agreement. Underwood et al. designed a mobile terrestrial scanning system for almond orchards that can efficiently map flower and fruit distributions and estimate the yield for individual trees [118]. The lidar canopy volume showed the strongest linear relationship to yield, with an $R^2$ of 0.77 for 39 tree samples spanning two years. Horton et al. developed an image processing algorithm to detect peach blossoms on trees [119]. Their method achieved an average detection rate of 84.3%, demonstrating its effectiveness in detecting peach blossoms. Inflorescence recognition and the detection of other fruits are shown in Table 10.

**Table 10.** Inflorescence recognition and the detection of other fruits.

| Fruits | Training Model/Method | Platform | Image | Performance Index | Reference |
|---|---|---|---|---|---|
| Pear | Yolov5s | Software platform: Ubuntu 20.04, Pytorch. Hardware platform: Intel(R)Core(TM)E5V3, CPU3.10 GHz, NVIDIA GeForce RTX3090. |  | Precision: 91.3% Recall: 89.9% Trained model size: 7.62 MB Detection speed: 34 FPS | [111] |
| Pear | Yolov4 | Software platform: Pytorch. |  | Accuracy of scale prediction: 94.89% Accuracy of density classification: 94.29% | [112] |
| Citrus | Yolov4 | Software platform: Windows 10, TensorFlow 2.2.0, Keras 2.3.1. Hardware platform: Intel(R)Core(TM) i7-9700, CPU3.00 GHz, NVIDIA GeForce RTX2060. |  | mAP: 95.03% F1 score: 89.00% Trained model size: 5.96 MB Detection speed: 30 FPS | [113] |
| Citrus | Mask R-CNN | Software platform: Python3, TensorFlow, Keras. Hardware platform: NVIDIA Quadro M4000 |  | AP value: 36.3 Average error rate: 11.9% | [114] |
| Grape | Mask R-CNN | Software platform: Windows 10, Pytorch. Hardware platform: Intel(R)Core(TM) i7, NVIDIA GeForce RTX2080Ti. |  | Precision: 83.3% Location time: 0.325 s | [115] |

**Table 10.** *Cont.*

| Fruits | Training Model/Method | Platform | Image | Performance Index | Reference |
|--------|----------------------|----------|-------|-------------------|-----------|
| Litchi | Yolov3, Yolov4, Faster-RCNN | Software platform: Ubuntu 16.04, PyTorch 1.3, Python 3.7. Hardware platform: Intel Xeon Gold 5218, CPU2.3 GHz, NVIDIA GeForce RTX2080Ti. |  | Mean absolute error: 16.29 Mean square error: 25.4 | [116] |
| Mango | Faster-RCNN | Software platform: Matlab. Hardware platform: Intel Core i7 CPU. |  | Precision: 89% Recall: 81% Accuracy: 96% F1 score: 0.85 | [117] |
| Almond | Point cloud with image colour | The "Shrimp" mobile ground vehicle robot |  | $R^2$ of hand-held photography and image processing: 0.71 | [118] |
| Peach | Multispectral | Unmanned aerial system with multispectral camera |  | Average detection rate: 84.3% | [119] |

## 5. Conclusions and Future Perspectives

Blossom thinning mechanization is an inevitable trend in the development of flower and fruit management in orchards. However, due to the constraints of canopy shapes and terrains, marketable mechanized blossom thinning equipment cannot be operated in the field, and the operation method is still in the stage of artificial thinning. Even in modern orchards that are suitable for mechanization, thinners are random hitting operations and cannot be finely tuned for different fruit trees. The phenomenon of leakage and excessive thinning often occurs, and the economic benefits are not maximized. In order to further enhance the mechanization level of orchard production management, promote the healthy development of the fruit industry, and comprehensively promote the construction of a smart fruit industry, research on orchard blossom thinning machinery can focus on the following aspects.

### 5.1. Orchard Planting Pattern

To achieve mechanization in orchard production management, the first step is to ensure that equipment can enter the orchard. The unity of orchard planting patterns is crucial, and consistent canopy shapes of fruit trees can ensure the smooth entry of blossom thinning arms into the interior of the canopy for operation. Strict planting standards

must be established for the spacing between ridges, rows, plants, and turning spaces to ensure the flexible movement of thinners. Existing old orchards need to be upgraded and renovated through methods such as transplanting, pruning branches, and traction to enable machinery to enter the orchard. When constructing a new orchard, it is necessary to fully consider the issue of mechanized operations and equip it with tractor roads and related supporting facilities. Orchard contiguous planting areas should be moderately scaled up for production and establish replicable and scalable cultivation models suitable for mechanized operations. Improving the level of mechanized production in orchards requires the integration of agricultural machinery and agronomy.

### 5.2. Mechanical Structure of the Thinning Arm

Different planting regions have different cultivation requirements. To facilitate mechanized operations, horticultural researchers have innovated various canopy shapes suitable for mechanization, such as the trunk type, the open center type, the Y trellis type, and the hedgerow type. However, the existing thinning equipment has a single mechanical structure with a single spindle. For non-hedgerow orchards, the rope cannot enter the canopy of the fruit trees effectively, resulting in uneven thinning densities and interference. For existing old orchards, the hand-held electric thinner can be used due to limited conditions. For existing mechanized orchards, profiling thinning arms can be developed for different tree shapes so that the thinning axis can be flexibly adjusted according to the fruit tree canopy to maximize the operation efficiency.

### 5.3. Material of the Thinning Rope

The principle of the mechanized thinning operation lies in the mutual beating between the rope and the inflorescence. During this process, the rope also interacts with the branches, as well as between the ropes and spindle, involving physical actions such as collision, friction, entanglement, compression, shearing, etc., which raise durability and flexibility issues for rope materials. Choosing a good material for the rope is a key consideration for thinner development. The ideal rope should have good flexibility without causing excessive damage to the fruit tree. The best materials for this purpose are plastics and rubber, which come in various forms, such as polyethylene, polypropylene, nylon, etc. Selecting or modifying the material formulation from existing materials to find a suitable rope material for thinning operations is a future research focus.

### 5.4. Parameterized Thinning Operation Model

Currently, mechanized thinning in orchards simply relies on the subjective judgment of operators to control the tractor speed, spindle rotational speed, thinning distance, rope density, etc., which is inconvenient and results in a low thinning precision. Existing research has not considered the differences in biological characteristics of branches and inflorescence during the blossom period, nor has it conducted in-depth tests on the flexible motion trajectory and beating force of the rope using branch models. The combination of a computer virtual simulation and indoor test bench test is insufficient to study the interaction mechanism between the rope and the inflorescence. The construction of a parameterized thinning operation model is crucial. 3D printing technology can be used to produce a physical model of fruit tree branches and construct a thinning test platform. The ADAMS virtual simulation and high-speed photography trajectory capture test can be combined to study the motion trajectory of the rope beating action and the beating force on branches and inflorescences. By integrating 3D printing technology, high-speed photography, and film pressure sensing technology, a parameterized thinning operation model can be built for orchards.

### 5.5. Intelligent Thinning

The number and position of inflorescences vary between individual fruit trees. Fixed machine operation parameters cannot guarantee the consistency of inflorescence densities

among each tree. A high inflorescence thinning rate will cause a reduced fruit production, while a low thinning rate will increase the workload of subsequent fruit thinning. Machine operation-based operator-subjective judgment cannot solve the above problems well. In recent years, deep learning neural networks have been extensively studied for flower recognition. By replacing human eyes with high-precision cameras and human brains with computer neural networks, fruit tree inflorescence multi-object recognition models based on deep learning algorithms such as Faster R-CNN, YOLO, and SSD have been developed, but most experimental studies are based on datasets and collect images at close distances. The future direction is to combine these with the thinner, in order to realize the recognition and positioning of the inflorescences in actual operations to achieve intelligent thinning.

**Author Contributions:** X.L. (Xiaohui Lei) and X.L. (Xiaolan Lyu) conceived the research idea; X.L. (Xiaohui Lei) wrote the paper; K.H., Y.Q., J.Z., and Y.S. processed the data; Q.Y., T.X., and A.H. reviewed and suggest the paper. All authors have read and agreed to the published version of the manuscript.

**Funding:** This research was funded by the National Natural Science Foundation of China (32201680), China Agriculture Research System of MOF and MARA (CARS-28-21), Jiangsu Modern Agricultural Machinery Equipment and Technology Demonstration Extension Fund (NJ2022-14), Jiangsu Agricultural Science and Technology Innovation Fund (CX(21)2025), National Key Research and Development Program of China (2022YFD2001400), National Science and Technology Development Program of China (NK2022160104), Jiangsu Policy-guided Plans (BX2019016), and Wuxi Science and technology development Fund (N20221003).

**Data Availability Statement:** Not applicable.

**Acknowledgments:** The authors would like to thank Xiaogang Li, Qingsong Yang, Zhonghua Wang, and Jialiang Kan (Institute of Pomology, Jiangsu Academy of Agricultural Science) for their help in orchard agronomy.

**Conflicts of Interest:** The authors declare no conflict of interest.

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
