# Peer review of "Technologies and Equipment of Mechanized Blossom Thinning in Orchards: A Review"

_agronomy, doi:10.3390/agronomy13112753_

Round 1

Reviewer 1 Report

Comments and Suggestions for Authors

This is a review paper on the design and applications of orchard mechanized blossom thinning equipment from three aspects: mechanism research, machine development, and intelligent upgrading. As described in the title of the paper: Technologies and equipment of mechanized blossom thinning in orchard. At present, there are few articles related to orchard mechanized blossom thinning equipment, and this paper has better innovation. The reviewer has some questions who want to ask, specifically as follows:

1. There are no associated pictures in Part 2, perhaps the author will be able to provide some pictures of the relevant content in the form of tables like subsequent tables.

2. Darwin series orchard single spindle string blossom thinner is the most common practical model on the market. What are their optimal working speed and the spindle rotation speed in apple orchard, peach orchard, etc.? Please indicate in paragraph 3 of Section 3.2.

3. Please indicate the research institutions of each model in Table 1, Table 2, Table 5, Table 7, and Table 8. This will help readers quickly find the research and development institution of the machine.

4. For Table 3, Table 4, Table 5, Table 6, and Table 7, it is suggested to add suitable tree species and canopy type in the main performance parameter column of each model.

Comments on the Quality of English Language

1. The English level of the article is average, and further improvement is recommended.

Author Response

Dear reviewer,

Thank you for your careful review. Your questions the author answers as follows:

1. Thinning force, thinning agronomy, and thinning model in part 2 are the results of some literal tests mainly. There is no classic representative picture, so the author does not provide pictures of the relevant content in the form of tables like subsequent tables.

2. The author has added the optimal working speed and the spindle rotation speed of Darwin series orchard single spindle string blossom thinner in paragraph 3 of Section 3.2. Please refer to the revised version for details.

3. The author has added the research institutions of each model in Table 1, Table 2, Table 5, Table 7, and Table 8. Please refer to the revised version for details.

4. The author has added the suitable tree species and canopy type in the main performance parameter in Table 3, Table 4, Table 5, Table 6, and Table 7. Please refer to the revised version for details.

5. The author has polished the revised version.

Thank you for your valuable review again.

Reviewer 2 Report

Comments and Suggestions for Authors

The paper is systematically written, the only thing that needs to be changed is the quality of the images.

Author Response

Dear reviewer,

Thank you for your careful review.

For the reason of the image quality, the author thinks is the format of PDF document. The document author uploaded in system is word (Size: 44.8 Mb), but the manuscript system gives you is PDF (Size: 1.6 Mb). Attached word document are the high quality images you point to, where the author changed Darwin SmaArt working site image of Table 3 in revised version.

Thank you for your valuable review again.

Reviewer 3 Report

Comments and Suggestions for Authors

1 It is necessary to indicate robotic platforms and technical means for plant thinning

2 It is advisable to present the research results in the form of a morphological matrix that determines the positive and negative aspects of the technical means under study.

3 An assessment of the technical means under study must be performed both in terms of the quality of work and the energy intensity of the work performed.

4 In addition, in my opinion, it is advisable to present, in addition to the tabular form of presenting the results, also in the form of graphs, reflecting the degree of implementation of the studied technical means in the real sector of the economy.

5 In the relevance of the presented manuscript, it is necessary to present the technological need for the technical means studied in the manuscript.

Author Response

Dear reviewer,

Thank you for your careful review. Your questions the author answers as follows:

1. The related contents of the blossom thinning robot system in this paper are mainly divided into two parts: one is the automatic thinning content in paragraph 7 of Section 3.2, and two is the intelligent thinning part in Section 4. Section 3.2 is a materialized automation innovation model, which is mainly to achieve the purpose of automatic thinning by adding sensors such as cameras, lidar and ultrasonic to the existing machine. According to the requirements of the review, the author adds the description of thinning mode and the test results of related models in paragraph 7 of Section 3.2. For intelligent thinning, the current relevant research only stays on the improvement of deep learning algorithms, and the technical means are not mentioned in the relevant references. In addition, the author has listed the basic model and hardware system adopted by the algorithm in Table 9, and listed the recognition effect.

2. The author has written the positive and negative aspects of the blossom thinning technical means under study, including 5 aspects: orchard planting pattern, mechanical structure of thinning arm, material of thinning rope, parameterized thinning operation model, and intelligent thinning.

3. In the aspect of thinning robots, the author has explained and corrected in question 1. In the aspect of mechanized thinning, for the hand-held thinner, the author stated the test results in paragraph 3 of Section 3.1. For Darwin series orchard single spindle string blossom thinner, the author stated the test results in paragraph 3 of Section 3.2. For Eclairvale series orchard thinner, the author stated the test results in paragraph 4 of Section 3.2. For three-arm blossom thinner, the author stated the test results in paragraph 5 of Section 3.2. For spiked-drum shaker, the author stated the test results in paragraph 6 of Section 3.2.

4. Due to the number of thinner models, the author inserts the typical working site pictures of each thinner into the table in the form of embedded.

5. The author has presented the blossom thinning technological need in paragraph 1, 2 of the Introduction.

Thank you for your valuable review again.